# Sauchinone Protects Renal Mesangial Cell Dysfunction against Angiotensin II by Improving Renal Fibrosis and Inflammation

**DOI:** 10.3390/ijms21197003

**Published:** 2020-09-23

**Authors:** Jung Joo Yoon, Hyeon Kyoung Lee, Hye Yoom Kim, Byung Hyuk Han, Ho Sub Lee, Yun Jung Lee, Dae Gill Kang

**Affiliations:** 1Hanbang Cardio-Renal Syndrome Research Center, Wonkwang University, 460, Iksan-daero, Iksan, Jeonbuk 54538, Korea; mora16@naver.com (J.J.Y.); gorud0170@naver.com (H.K.L.); hyeyoomc@naver.com (H.Y.K.); arum0924@naver.com (B.H.H.); host@wku.ac.kr (H.S.L.); 2College of Oriental Medicine and Professional Graduate School of Oriental Medicine, Wonkwang University, 460, Iksan-daero, Iksan, Jeonbuk 54538, Korea

**Keywords:** sauchinone, angiotensin II, mesangial cell, fibrosis, inflammation

## Abstract

Abnormal and excessive growth of mesangial cells is important in the pathophysiologic processes of diabetes-associated interstitial fibrosis and glomerulosclerosis, leading to diabetic nephropathy, which eventually turns into end-stage renal disease. Sauchinone, a biologically-active lignan isolated from aerial parts of *Saururus chinensis*, has anti-inflammatory and anti-viral activities effects on various cell types. However, there are no studies reporting the effects of sauchinone on diabetic nephropathy. The present study aims to investigate the role of sauchinone in mesangial cell proliferation and fibrosis induced by angiotensin II, as well as the underlying mechanisms of these processes. Human renal mesangial cells were induced by angiotensin II (AngII, 10 μM) in the presence or absence of sauchinone (0.1–1 μM) and incubated for 48 h. In this study, we found that AngII induced mesangial cell proliferation, while treatment with sauchinone inhibited the cell proliferation in a dose-dependent manner. Pre-treatment with sauchinone induced down-regulation of cyclins/CDKs and up-regulation of CDK inhibitor, p21, and p27^kip1^ expression. In addition, AngII-enhanced expression of fibrosis biomarkers such as fibronectin, collagen IV, and connective tissue growth factor (CTGF), which was markedly attenuated by sauchinone. Sauchinone also decreased AngII-induced TGF-β1 and Smad-2, Smad-3, and Smad-4 expression. This study further revealed that sauchinone ameliorated AngII-induced mesangial inflammation through disturbing activation of inflammatory factors, and NLRP3 inflammasome, which is composed of the NLRP3 protein, procaspase-1, and apoptosis-associated speck-like protein containing a CARD (ASC). Moreover, pretreatment of sauchinone inhibited NF-κB translocation and ROS production in AngII-exposed mesangial cells. These data suggest that sauchinone has a protective effect on renal proliferation, fibrosis and inflammation. Therefore, sauchinone might be a potential pharmacological agent in prevention of AngII-induced renal damage leading to diabetic nephropathy.

## 1. Introduction

Diabetic nephropathy (DN) is a representative complication of diabetes and is a leading cause of end-stage renal disease [1]. DN is characterized by aberrant alterations such as glomerular mesangial cell proliferation, hypertrophy, and extracellular matrix (ECM) accumulation in the glomeruli [2,3]. The first and most prominent glomerular lesion of diabetic nephropathy is mesangial expansion concurrently accompanying mesangial hyperplasia called diabetic glomerulosclerosis. An imbalance in the control of mesangial cell proliferation appears to play an early and crucial role in progressive glomerular injury that leads to glomerular sclerosis [4]. Currently, mesangial cells are recognized as the major cells to secrete ECM [5]. Therefore, inhibiting the proliferation of mesangial cells and the accumulation of ECM can be applied as a practical approach for treating or relieving DN [6]. However, the underlying mechanism is still unsatisfactory, and there is no effective therapeutic treatment drug.

Angiotensin II (AngII), a major component in the renin–angiotensin system (RAS), plays a pivotal role and is not only activated systemically in diabetic nephropathy, but also locally activated in each organ [7]. AngII is known to significantly contribute to the induction of renal damage through the glomerular mesangial expansion, glomerular basement membrane glomerular basement membrane splitting, and ECM deposition [8]. AngII has been reported to cause progressive renal dysfunction by inducing the contraction of the glomerular mesangial cells, which contributes to the development of glomerular sclerosis. Numerous studies have demonstrated that chronic exposure to AngII may also result in detrimental effects on the kidney through cellular mechanisms that stimulate mesangial cell proliferation and fibrotic factors [9].

Transforming growth factor-β1 (TGF-β1) is a key mediator of glomerulosclerosis and fibrosis leading to end stage renal disease. TGF-β activation strongly causes increased ECM synthesis, mesangial proliferation, and glomerular fibrosis in DN renal tissue [10]. Binding of TGF-β to its receptor leads to phosphorylation of the down-stream receptor-associated Smads (R-Smads) i.e., Smad2 and Smad3 [11,12]. Then the phosphorylated Smad2/3 forms an oligomeric complex with a common Smad (C-Smad), Smad4, and translocates into the nucleus leading to increases in the expression and synthesis of major collagenous components of ECM such as connective tissue growth factor (CTGF), type IV collagen (ColIV), and fibronectin [13,14]. Therefore, the TGF-β pathway plays an important role in renal fibrosis and is a critical target for glomerular therapies. The main ECM proteins such as ColIV and fibronectin are often used as the markers of fibrogenesis in many kidney fibrosis diseases including diabetic glomerulosclerosis [15,16]. CTGF, a downstream molecule of TGF-β/Smad, is a novel profibrotic factor that is up-regulated in the glomerulosclerosis of DN. Recent studies have identified that CTGF promotes the abnormal deposition of ECM and fibrosis and thus could enhance the disassembly and hypertrophy of mesangial cells [17].

Renal inflammation plays a major contributor in the pathogenesis of diabetic nephropathy because inflammation can induce fibronectin expression and ECM accumulation, and consequently accelerate the progress of glomerulosclerosis and tubulointerstitial fibrosis [18]. Recent studies revealed that monocyte chemoattractant protein-1 (MCP-1) and intercellular adhesion molecule-1 (ICAM-1) are overexpressed in the glomeruli from diabetic animals and has been reported to be associated with development of diabetic nephropathy [19]. Inflammatory chemokine MCP-1 plays an important role in the pathogenesis of progressive glomerular and tubulointerstitial lesions in different animal models of renal damage and human renal diseases. ICAM-1 is the most important adhesion molecule in the inflammatory response process and it is thus a potential target of MCP-1 action [20]. Nuclear factor-κB (NF-κB) is a transcription factor widely distributed in most cell types, including glomerular mesangial cells and its signaling pathways also play a key role in tissue inflammation and cell apoptosis [21]. In mesangial cells, activation of NF-κB can be induced by various molecules such as cytokines and reactive oxygen species (ROS) and it controls the transcription of several genes involved in immune and inflammatory responses, cell growth, and adhesion [22].

Several studies have reported that ROS play an important role in the pathogenesis of renal profibrotic factors in inducing fibroblast proliferation [23] and may directly or indirectly activate the NOD-like receptor family, pyrin domain-containing-3 (NLRP3) inflammasome. The NLRP3 inflammasome is composed of NLRP3, apoptosis-associated speck-like protein containing a CARD (ASC), and Caspase-1 [24], and it has been reported that inhibiting renal NLRP3 activation could significantly reduce renal inflammation and improve the renal functions involved in the development of chronic kidney diseases [25,26]. In addition, NLRP3 inflammasome activates caspase-1 which can promote the maturation and secretion of inflammatory cytokine interleukin 1β (IL-1β) and trigger a powerful and endogenous inflammatory cascade reaction, which would cause the occurrence and development of DN [27].

*Saururus chinensis*, a perennial herb commonly called Sam-baekcho in Korea, has been traditionally used in the treatment of edema, gonorrhea, jaundice, and inflammatory diseases. Among *Saururus chinensis* active compounds, sauchinone, one of biologically-active lignans, has been reported to show a variety of bioactivities such as anti-oxidant, anti-inflammatory, anti-apoptosis, and anti-obesity effects in several cell types [28,29,30,31]. Previous studies have shown that sauchinone protected against the production of lipopolysaccharide (LPS)-induced inflammatory factors such as nitric oxide [32]. However, the effect of sauchinone on diabetic nephropathy is still unclear. In the present study, we investigated the protective effect of sauchinone against AngII-induced renal dysfunction such as renal fibrosis and inflammation in mesangial cells.

## 2. Results

### 2.1. Effect of Sauchinone on Mesangial Cell Proliferation

As shown in Figure 1, AngII showed significantly-greater proliferation compared to the control. The pretreatment with sauchinone inhibited Ang II-induced cell proliferation in a dosage-dependent manner (Figure 1A). AngII-induced increase of cell numbers also reduced by adding more than 0.5 μM concentration of sauchinone (Figure 1B). Next, using the iCELLigence microelectronic biosensor system, the real-time analysis of proliferation of established cell cultures was performed and cell proliferation was monitored in real time. We observed that cell index increase started to decrease after about 24 h of incubation with sauchinone. Pretreatment with sauchinone suppressed cell proliferation after 48 h of incubation, whereas AngII increased cell index up to point in that time (Figure 1C,D).

To further evaluate the effect of sauchinone on mesangial cell proliferation, we used wound-healing assays and Western blot analysis. The result revealed a significant decrease in the wound-healing distance by AngII after 48 h. However, the wound-healing distance of the cells pretreated with sauchinone was improved compared to the AngII (Figure 2A). Cyclins and CDKs are a key gene in cell cycle regulation which controls the G1-S transition during the cell cycle. As shown in Figure 2B,C, AngII induced the expression of CyclinE, Cyclin D1, CDK2, and CDK4 protein expression and decreased levels of p21^waf1/cip1^ and p27^kip1^, while those of the cell cycle-related proteins were restored due to the pretreatment with sauchinone. These results indicated that sauchinone inhibited AngII-promoted mesangial cell proliferation by inhibiting G1-S phase transition and arresting cells in G0/G1.

### 2.2. Effect of Sauchinone on Mesangial Cell Fibrosis

The effect of sauchinone on AngII-induced fibrosis was evaluated by determining the protein and mRNA expression of renal fibrosis related proteins. As shown in Figure 3, AngII significantly increased the protein expression of fibronectin, ColIV, and CTGF, which was remarkably decreased by adding ≥ 0.5 μM sauchinone pretreatment. Pretreatment with sauchinone also reduced AngII-induced fibronectin and ColIV mRNA levels (Figure 3B, *p* < 0.01). To further evaluate the effect of sauchinone on AngII-induced mesangial cell fibrosis, we next performed Western blot and RT-qPCR assay for TGF-β/Smads signaling. As a result, AngII-enhanced cellular levels of TGF-β, p-Smad-2, p-Smad-3, and Smad-4 at 48 h after stimulation, while sauchinone decreased protein expression of TGF-β/Smad pathway (Figure 3C). The results of TGF-β, p-Smad-2, and p-Smad-3 mRNA expression experiments using RT-PCR were similar to those of protein expression (Figure 3D). In addition, as shown in Figure 3E, staining intensities of Smad-2 in the nucleus highly increased under the AngII condition; however, Smad-2 expression levels markedly decreased when pretreated with sauchinone. The findings demonstrated that sauchinone could inhibit AngII-induced renal fibrosis through inhibiting synthesis of ECM proteins and blocking TGF-β-mediated pathway in mesangial cells.

### 2.3. Effect of Sauchinone on Mesangial Cell Inflammation

To evaluate the role of sauchinone on mesangial cell inflammation, cells were pre-treated with sauchinone before AngII stimulation. The Western blot results revealed that ICAM-1 and MCP-1 expressions were all significantly increased in the AngII condition compared with the control. However, these increases were decreased by pretreated with sauchinone (Figure 4A). RT-PCR analysis revealed that ≥0.5 μM sauchinone exerted an inhibitory effect on AngII-increased ICAM-1 and MCP-1 mRNA expression (Figure 4B). To investigate the effect of sauchinone on AngII-induced expression of NLRP3 inflammasomes in renal mesangial cells, Western blotting was used to assess the expression of NLRP3 inflammasome. As shown in Figure 5A, AngII up-regulated the protein expression of NLRP3, ASC, caspase-1, and IL-1β. In addition, NLRP3, ASC, and caspase-1 mRNA levels were enhanced in AngII-exposed mesangial cells, evidenced by RT-PCR and real-time PCR. This elevation was markedly attenuated by treated by ≥0.5 μg/mL sauchinone (Figure 5B). These results demonstrate that sauchinone may alleviate mesangial inflammation possibly involved in renal fibrotic process by mediating the NLRP3-caspase-1-IL-1β/IL-18 signaling pathway.

### 2.4. Effect of Sauchinone on Inflammatory Responses by NF-κB/ROS Pathway

To investigate whether sauchinone initiates a cellular inflammatory response via the NF-κB pathway, we assessed the NF-κB/ROS signaling pathway. Western blot results indicated that Ang II increased IκB-α phosphorylation and nuclear distribution of NF-κB p65 (one of the NF-kB subunit proteins). Sauchinone effectively prevented these Ang II-mediated effects in mesangial cell (Figure 6A). To identify the localization of NF-κB p65, we performed immunofluorescence analysis. However, the Ang II-induced nuclear translocation of NF-κB p65 was effectively inhibited by pretreated with ≥0.5 μM sauchinone (Figure 6B). We tested the effects of sauchinone on AngII-induced intracellular ROS generation by dichlorofluorescin diacetate (DCFH-DA) fluorescent assay using fluorescence microscopy. As a result, the cells cultured under AngII conditions for 48 h were observed to increase in ROS generation compared with control. However, when the cells were pretreated with sauchinone, the Ang II-induced translocation of NF-κB p65 was inhibited (Figure 6C,D). These data suggest that sauchinone inhibits renal inflammation, probably by preventing NF-κB/ROS pathway activation.

## 3. Discussion

Renal mesangial cells, one of the major types of resident renal cells, play important roles in various kidney diseases, including diabetic nephropathy. Abnormal and excessive growth of mesangial cells is the main cause of interstitial fibrosis and renal function loss, resulting glomerulosclerosis, which eventually turns into end-stage renal disease [33]. DN, a well-known microvascular complication in patients with diabetes and a common cause of end-stage renal disease worldwide, is characterized by glomerular hypertrophy, the accumulation of extracellular matrix protein, which leads to glomerular fibrosis [2]. Mesangial cells are not the main cause of interstitial fibrosis, but definitely contribute to glomerular sclerosis.

High glucose and Ang II have been shown to produce similar effects on renal cells in culture. For instance, incubation of mesangial cells in high-glucose media or in the presence of Ang II stimulates matrix protein synthesis and inhibits degradative enzyme activity [7]. In addition, numerous studies have demonstrated that the intrarenal renin–angiotensin system plays an important role in diabetic nephropathy. AngII, a major component in the RAS, is known to significantly contribute to the induction of renal damage through the glomerular mesangial expansion, glomerular basement membrane glomerular basement membrane splitting, and ECM deposition [7,8]. Although current treatment can delay the progression of DN, it is still limited. This research proved for the first time that sauchinone isolated from the aerial part of *S. chinensis* improves diabetic nephropathy by alleviating AngII-induced glomerular fibrosis and inflammation in human kidney mesangial cells.

Mesangial cells migration and proliferation are fundamental responses to glomerular injury and are observed in a number of glomerular diseases that progress to glomerular sclerosis [34]. The present study demonstrated that AngII induced marked increases in mesangial cell proliferation and migration. Stimulation of mesangial cell proliferation by AngII has already been demonstrated in previous studies [35]. In the present study, [^3^H]-thymidine incorporation and mesangial cell index were measured to confirm that sauchinone inhibited AngII-induced mesangial cell proliferation. The results of the present study demonstrated sauchinone’s improvement effect on aberrant proliferation of mesangial cells through control of cell cycle regulatory proteins such as CDK complexes (cyclin D1/CDK-4 and cyclin E/CDK-2) and CDK inhibitors (p21^waf1/cip1^ and p27^kip1^). In addition, based on the results of wound-healing assay, we confirmed that sauchinone significantly inhibited mesangial cell migration. These data suggested that sauchinone might be useful in preventing the glomerular diseases that progress to glomerular sclerosis.

Glomerular mesangial ECM mainly consists of ColIV, fibronectin, laminins, and HSPGs. They affect renal cell adhesion, growth, migration, and proliferation [36]. More recently, CTGF has been suggested as a major cytokine responsible for elevated deposition of ECM proteins, and is known to be strongly upregulated in diabetic nephropathy and other advanced kidney diseases [37]. In addition, TGF-β is known as a key pro-fibrotic mediator in the development of kidney diseases including renal fibrosis and it is well documented that TGF-β1 has multiple biological properties including cell proliferation, differentiation, apoptosis, and production of ECM. [12,38]. The results of the present study suggested that fibronectin, ColIV, and CTGF may be involved in the ECM-synthesizing process, and that their AngII-induced protein and mRNA expression were decreased following treatment with sauchinone. We recently reported that HG or AngII induces mesangial cell fibrosis by activation of TGF-β/Smad signaling [39]. We have now shown that sauchinone inhibited the elevated expression of TGF-β and Smad2/3 phosphorylation under AngII conditions. These findings have demonstrated that sauchinone may have a significant protective effect against AngII-induced renal dysfunction by control of pro-fibrotic mediators and TGF-β signaling pathways.

Various theories have been proposed concerning the pathogenesis of diabetic nephropathy, including proteinuria, genetics, hypoxia, ischemia, and inflammation. Among these theories, inflammation has previously been reported as an important pathway for the development and progression of diabetic nephropathy [40,41]. Thus, controlling excessive inflammation has great therapeutic potential to suppress diabetic nephropathy caused by progressive renal fibrosis. In our studies, ICAM-1 and MCP-1 expression were all significantly increased in the AngII condition compared with the control. However, these increases were decreased by pretreatment of sauchinone. In addition, AngII upregulated the expression of NLRP3, ASC, caspase-1, and IL-1β. This elevation was markedly attenuated by sauchinone. These results demonstrate that sauchinone may alleviate mesangial inflammation possibly involved in renal fibrotic process by mediating the NLRP3–caspase-1 signaling pathway. Additionally, the NF-κB/ROS pathway is involved in the transcriptional regulation of hundreds of genes related to inflammation, immunity, apoptosis, cell proliferation, and differentiation. The activated NF-κB induce target genes of NF-κB, including ICAM-1 and MCP-1, which in turn enhance inflammation and finally lead to the acceleration of the pathogenesis of glomerulosclerosis and renal fibrosis through ECM accumulation [42]. Our results shown that AngII stimulation increased expression of NF–kB and activated ROS production. However, sauchinone down-regulated transcriptional responses of nuclear NF-κB proteins and up-regulated phosphorylating I-κB-α. Sauchinone also significantly suppressed AngII-induced ROS production. Results from the present study suggest that sauchinone may control renal inflammation by regulating the expression of inflammation-related genes and the NF-κB/ROS signaling pathway. Additionally, this study further revealed that sauchinone ameliorated AngII-induced mesangial inflammation through disturbing activation of inflammatory factors, and NLRP3 inflammasome composed of the NLRP3 protein, procaspase-1, and ASC. However, two major limitations of the present study need to be acknowledged and addressed. First, we attempted to demonstrate the protective effect on diabetic nephropathy caused by progressive renal fibrosis through inflammation control of sauchinone by performing the study on a cell but not on a system that is more relevant to an in vivo situation. Second, in this study, the effect on mesangial cell by pre-treatment of sauchinone was studied. However, post-treatment research is also necessary to clarify the effects of sauchinone after the onset of the disease. More experiments, such as the use of animal models and post-treatment of sauchinone, should be performed in future studies to confirm the effect of sauchinone in improving diabetic nephropathy.

These data suggest that sauchinone has a protective effect on renal dysfunction caused by progressive renal fibrosis through inflammation control. Thus, sauchinone may be a potential treatment or prevention agent that can improve the renal dysfunction, which may develop into diabetic nephropathy.

## 4. Materials and Methods

### 4.1. Chemicals

Angiotensin II (10 μM, A9525) and sauchinone (SML0783) was purchased from Sigma-Aldrich (St. Louis, MO, USA). Alexa Four 488 phalloidin, dichlorofluorescin diacetate (DCFH-DA), DAPI, primary antibodies for p-Smad-2 (44-244G), p-Smad3 (44-246G), and NLRP3 (MA5-23919) were purchased from Thermo Fisher Scientific (Waltham, MA, USA). Primary antibodies for CDK2 (SC-6248), CDK4 (SC-23896), cyclinD1 (SC-8396), cyclinE (SC-247), p21 (SC-397), p27 (SC-1641), Smad-4 (SC-7966), CTGF (SC-373936), ICAM-1 (SC-8439), NfκB p65 (SC-8008), p-IκB-α (SC-8404), ASC (SC-514414), and IL-1β (SC-12742) were purchased from Santa Cruz Biotechnology (Santa Cruz, CA, USA). MCP-1 (44-246G) was purchased from Abcam (Cambridge, MA, USA). Capase-1 (4199) and HRP conjugated secondary antibodies raised against mouse, rabbit, and goat were purchased from Cell Signaling Technology (Danvers, MA, USA).

### 4.2. Cell Cultures

Primary human renal mesangial cells were purchased from ScienCell Corporation (Carlsbad, CA, USA) and cultured in low-glucose Dulbecco’s modified Eagle’s medium (DMEM; Gibco, Grand Island, NY, USA) supplemented with 10% fetal bovine serum (Gibco, Grand Island, NY, USA) and 1% antibiotic–antimycotic (Gibco, Grand Island, NY, USA). The dispersed mesangial cells were incubated in a humidified incubator at 37 °C under 95% air and 5% CO_2_.

### 4.3. Measurement of Cell Proliferation

[^3^H]-thymidine incorporation was measured to determine the effect on human mesangial cell proliferation. The mesangial cells were plated at 2.5 × 10^4^ cells/well in 24-well cell culture plates for 24 h, and the medium was replaced with serum-free medium containing 0.1% bovine serum albumin and the incubation continued for another 24 h. Quiescent cells were treated with 10 μM angiotensin II and DS-EA, respectively, and 1 µCi of [^3^H]-thymidine was added (methyl-[^3^H] thymidine 50 Ci/mmol; Amersham, Oakville, ON, Canada). After incubation for 24 h, cells were extracted three times with cold 10% TCA for 5 min each time and solubilized for at least 30 min in 0.3 N NaOH. After neutralizing, [^3^H]-thymidine activity was measured in a liquid scintillation counter (Beckman LS 7500, Fullerton, CA, USA). Each experiment was performed in triplicate or quadruplicate. Mesangial cell index was calculated for each E-plate well by RTCA Software 1.2 (Roche Diagnosis, Meylan, France). The graphs are real-time generated outputs from the iCELLigence system.

### 4.4. Wound Healing Assay, Migration Assay

The cells (1 × 10^5^) were incubated in 6-well culture plates. After being grown to 80–90% confluence, cells were scratched using a 200 μL yellow pipette tip. Images were captured by a microscope (Axiovision 4, Zeiss, Oberkochen, Germany) at 0 and 48 h. In each group, three duplicate wells were assayed, and each assay was conducted at least three times (magnification, ×100).

### 4.5. Western Blot Analysis

The mesangial cells (1 × 10^7^/dish) were seeded in 100 mm petri dish, and then treated with sauchinone with or without AngII for 48 h. Cell homogenates (40 μg of protein) were separated on 10% SDS-polyacrylamide gel electrophoresis and transferred to nitrocellulose paper. Blots were then washed with H_2_O, blocked with 5% skimmed milk powder in TBST [10 mM Tris-HCl (pH 7.6), 150 mM NaCl, 0.05% Tween-20] for 1 h and incubated with the appropriate primary antibody at dilutions recommended by the supplier. Then the membrane was washed, and primary antibodies were detected with goat anti-rabbit-IgG conjugated to horseradish peroxidase, and the bands were visualized with enhanced chemiluminescence (Amersham, Buckinghamshire, UK). Protein expression levels were determined by analyzing the signals captured on the nitrocellulose membranes using the Chemi-doc image analyzer (Bio-Rad, Hercules, CA, USA).

### 4.6. Preparation of Cytoplasmic and Nuclear Extracts

The renal mesangial cells were rapidly harvested in cold PBS on ice by sedimentation and centrifuged at 13,000 rpm for 10 min at 4 °C. Nuclear and cytoplasmic proteins were extracted with Nuclear Extract Kit (Active Motif, Inc., Carlsbad, CA, USA). Nuclear and cytosolic protein extracts were immediately transferred to a clean pre-chilled tube and all extracts stored at −80 °C until use.

### 4.7. RNA Isolation and Real-Time qRT-PCR

In real time PCR, well-grown cells in DMEM medium were plated in 6-well plates to 1 × 10^5^ cell/well. A kit from Qiagene (RNeasy™ Plus mini kit, QIAGEN, Hilden, Germany) was used for RNA isolation from cell cultures, and RNA quality was tested by measuring the ratio 260/280 nm in a UV-spectrophotometer. Real-time quantitative RT-PCR analysis was carried out in a 48-well plate using the Opticon MJ Research instrument (Bio-rad Inc., Hercules, CA, USA) and optimized standard SYBR Green 2-step qRT-PCR kit protocol (DyNAmo™, Finnzymes, Finland). Specific sense and antisense primers used were as follows, respectively: TGF-β1 (156 bp), sense: 5′-CCC AGC ATC TGC AAA GCT C-3′, anti-sense: 5′-GTC AAT GTA CAG CTG CCG CA-3′; Collagen IV (177 bp), sense: 5′-GGT GTT GCA GGA GTG CCA G-3′, anti-sense: 5′-GCA AGT CGA AAT AAA ACT CAC CAG-3′; Smad-2 (198 bp), sense: 5′-GTT CCT GCC TTT GCT GAG AC-3′, anti-sense: 5′-TCT CTT TGC CAG GAA TGC TT-3′; Smad-3 (183 bp) sense: 5′-GTGCTCCATCTCCTACTAC-3, anti-sense: 5′-CCTCCTCCGATGTGTCTC-3′; ICAM-1 (121 bp), sense: 5′-GCT GCT ACC ACA CTG ATG ACG ACA A-3, anti-sense: 5′-CAG TGA CCA TCT ACA GCT TTC CGG-3′; MCP-1 (185 bp), sense: 5′-GAT CTC AGT GCA GAG GCT CG-3′, anti-sense: 5′-TGC TTG TCC AGG TGG TCC AT-3′; NLRP3 (116 bp), sense: 5′-AGA GCC TAC AGT TGG TGA AAA TG-3, anti-sense: 5′-CCA CGC CTA CCA GGA AAT CTC-3′; GAPDH (86 bp), sense: 5′-CGA GAA TGG GAA GCT TGT CAT C-3′, anti-sense: 5′-CGG CCT CAC CCC ATT TG-3′. The PCR was started at 95 °C for 15 min (hot start) to activate the AmpliTaq polymerase, followed by a 45-cycle amplification (denaturation at 94 °C for 20 s, annealing at 60 °C for 30 s, extension at 72 °C for 60 s, and plate reading at 60 °C for 10 s). The temperature of PCR products was elevated from 65 °C to 95 °C at a rate of 0.2 °C/1 s, and the resulting data were analyzed by using the software provided by the manufacturer.

### 4.8. Immunofluorescence Microscopy

The mesangial cells were seeded into 6-well plates at a density of 2.5 × 10^5^ cell/well. Rat mesangial cells on glass coverslips were fixed in 4% paraformaldehyde (PFA) for 30 min and then permeabilized with 0.4% Triton X-100 for 5 min in PBS, washed 3 times with PBS. After blocking in 1% BSA, samples were incubated with primary antibody (p-Smad-2, Smad-4, and NF-κB p65) at 4 °C overnight. Corresponding secondary antibodies were labeled with Alexa Fluor 488 (1:200; Molecular Probes, Eugene, OR, USA). Rat mesangial cell nuclei were counterstained with DAPI. Recording and analysis of fluorescence signals were performed using ImagePro software 5.0 (Media Cybernetics, Inc., Rockville, MD, USA).

### 4.9. Measurement of ROS

The fluorescent probe CM-H_2_DCFDA was used to measure the intracellular generation of ROS. Briefly, the cells were seeded into 96-well plates at a density of 3 × 10^3^ cells/well and the confluent cell in the 96-well culture plates were pretreated with sauchinone for 30 min and stimulated in absence or presence of AngII for 48 h. The cells were incubated at 37 ℃ with 10 µM CM-H_2_DCFDA. Fluorescence intensity was measured by fluorescence microscopy (Infinite F200 pro, TECAN (Hombrechtikon, Switzerland): excitation 488 nm, emission 513 nm) and examined under a fluorescence microscope (Eclipse Ti, Nikon, Tokyo, Japan).

### 4.10. Statistical Analysis

All the experiments were repeated at least three times. The results were expressed as a mean ± S.E., and the data were analyzed using one-way ANOVA followed by a Dunnett’s test or Student’s *t*-test to determine any significant differences. *p* < 0.05 was considered as statistically significance.

## 5. Conclusions

The present study demonstrated that sauchinone has a beneficial effect on diabetic nephropathy by alleviating the glomerular fibrosis and improving inflammation. Thus, sauchinone might be a potential pharmacological agent in prevention of AngII-induced renal damage leading to diabetic nephropathy.

## Figures and Tables

**Figure 1 ijms-21-07003-f001:**
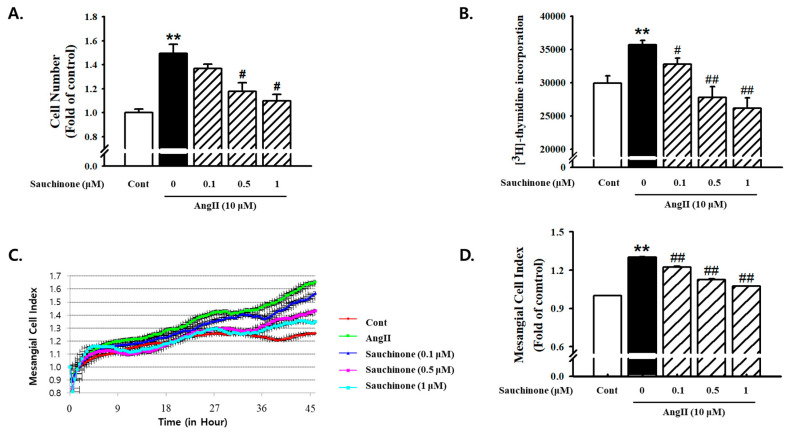
Effects of sauchinone on renal mesangial cell proliferation. Human mesangial cells were treated with angiotensin II (AngII) (10 μM) and incubated with or without sauchinone (from 0, 0.1, and 1 μM) for 48 h. (**A**) Cell number results of mesangial cells using Countess™ cell counting, (**B**) cell proliferation results using [^3^H]-thymidine incorporation assay, (**C**) results of mesangial cell index using xCELLigence RTCA DP Real time cell analyzer, and (**D**) quantification results of mesangial cell Index (fold of control). Results are expressed as the mean ± S.E. from four independent experiments. ** *p* < 0.01 vs. control; # *p* < 0.05, and ## *p* < 0.01 vs. AngII alone.

**Figure 2 ijms-21-07003-f002:**
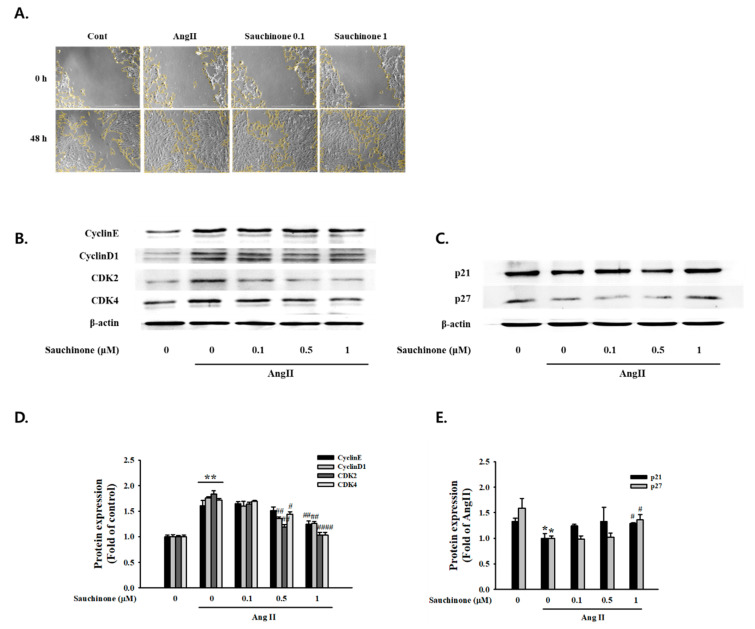
Effect of sauchinone on AngII-induced mesangial cell invasion and cell cycle-related proteins. (**A**) Microscopy images of the wound-healing assay performed on mesangial cells at 0 h and 48 h. Each photograph is representative of at least three independent experiments (magnification: X200). (**B**,**C**) Effects of sauchinone on expression of G0/G1 phase cell cycle-related proteins in mesangial cells. Expression of cell cycle-related proteins in mesangial cells was determined by Western blot analysis. (**D**,**E**) Quantitative results demonstrating that sauchinone regulated the cell cycle-related proteins in response to AngII. Each value represents the means ± S.E. of five independent experiments. ** *p* < 0.01, * *p* < 0.05 vs. control; ## *p* < 0.01, and # *p* < 0.05 vs. AngII alone.

**Figure 3 ijms-21-07003-f003:**
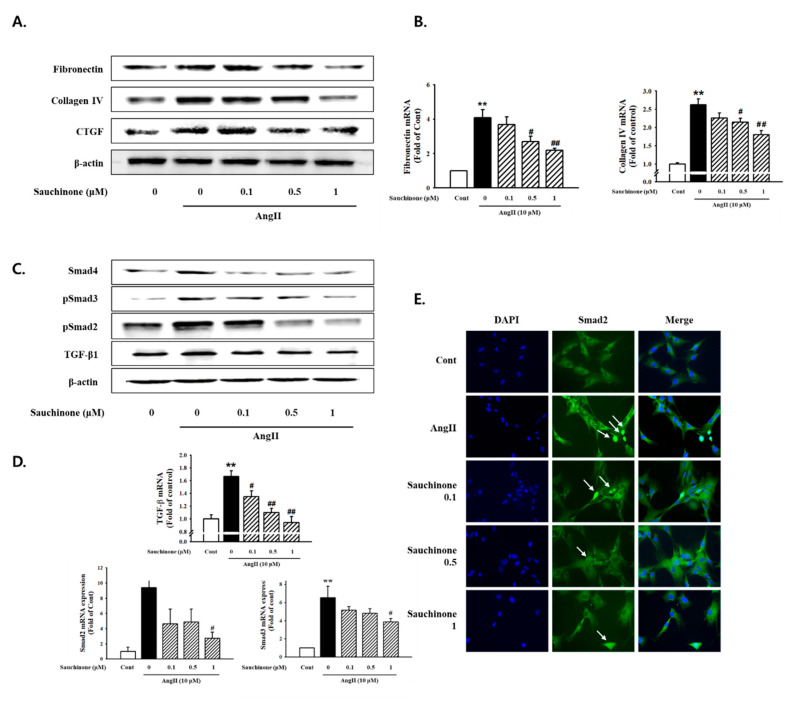
Effect of sauchinone on AngII-induced mesangial cell fibrosis. (**A**) Western blotting and (**B**) real-time PCR showed protein and mRNA levels of fibronectin, Collagen IV, and CTGF in sauchinone-treated and AngII-stimulated cells at 48 h. (**C**,**D**) Effect of sauchinone on the relative levels of TGF-β1/Smads. The results were detected by Western blotting and real-time PCR. (**E**) Immunofluorescent images of p-Smad-2 nuclear translocation under the laser scanning confocal microscopy are show (magnification. 400×). Nuclei were stained with DAPI (blue) and p-smad-2 was stained with Alexa Fluor 488 (green). Green fluorescence indicates localization of p-Smad-2. Respective blot data were obtained from five independent experiments. Each value represents the means ± S.E. of five independent experiments. ** *p* < 0.01 vs. control and ## *p* < 0.01, # *p* < 0.05 vs. AngII alone.

**Figure 4 ijms-21-07003-f004:**
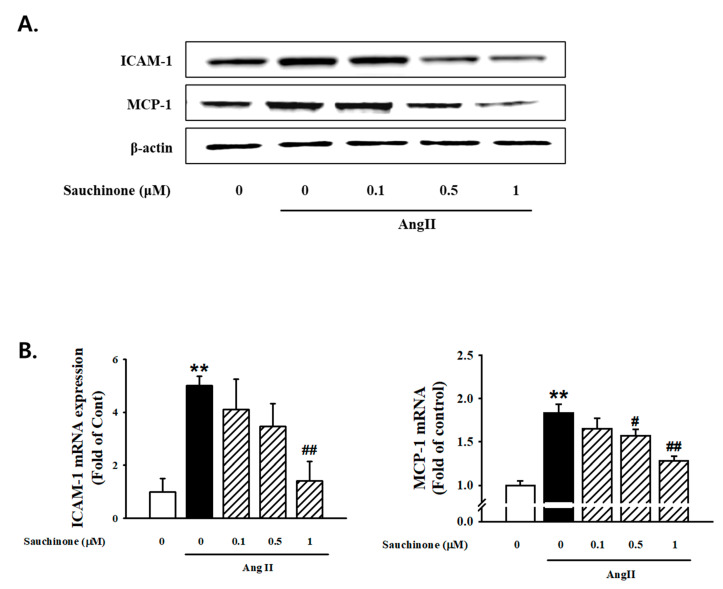
Effect of sauchinone on AngII-induced mesangial cell inflammation. (**A**) Western blotting and (**B**) real-time PCR showed protein and mRNA levels of intercellular adhesion molecule-1 (ICAM-1) and monocyte chemoattractant protein-1 (MCP-1) in sauchinone-treated and AngII-stimulated cells at 48 h. β-actin or GAPDH were used as the internal standard in each sample. ** *p* < 0.01 vs. control and ## *p* < 0.01, # *p* < 0.05 vs. AngII alone.

**Figure 5 ijms-21-07003-f005:**
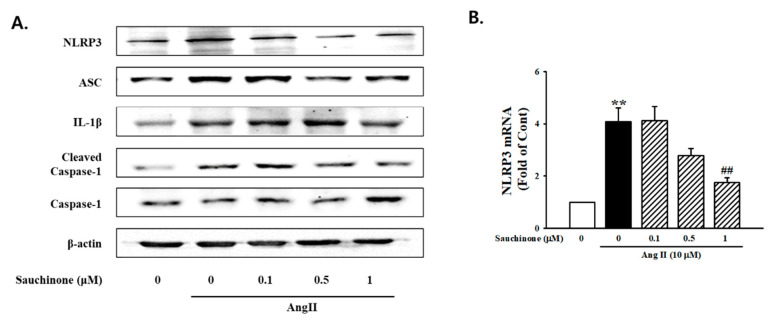
Inhibition effects of sauchinone on the NOD-like receptor family, pyrin domain-containing-3 (NLRP3) inflammasome in mesangial cells. (**A**) Western blotting and (**B**) real-time PCR showed protein and mRNA levels of NLRP3, ASC, IL-1β, cleaved caspase-1, and caspase-1 in sauchinone-treated and AngII-stimulated cells at 48 h. β-actin or GAPDH were used as the internal standard in each sample. ** *p* < 0.01 vs. control and ## *p* < 0.01 vs. AngII alone.

**Figure 6 ijms-21-07003-f006:**
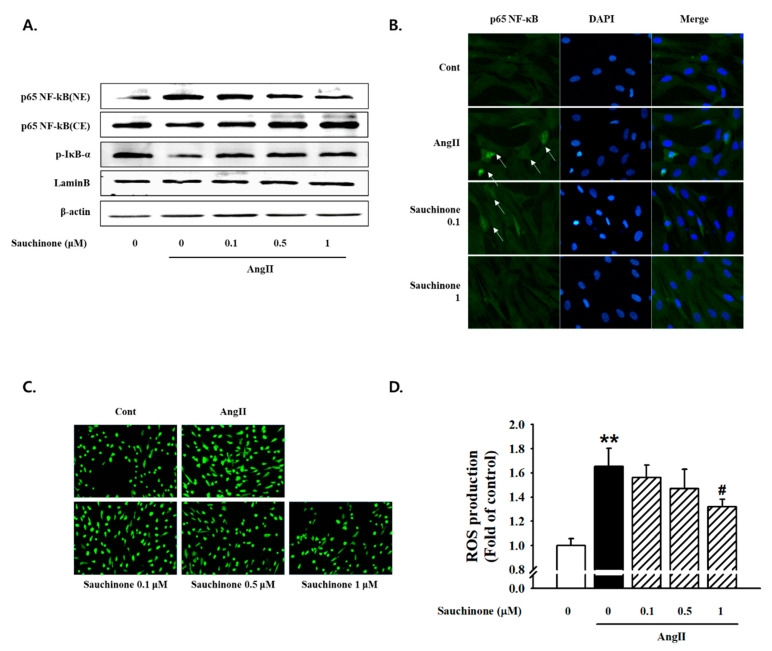
Effect of sauchinone on nuclear factor-κB (NF-κB)/ROS pathway. (**A**) Blockade of nuclear translocation of NF-κB in mesangial cells. For Western blot analysis with a primary antibody against NF-κB/p-IκB-α, cytoplasmic (CE) and nuclear (NE) fractions were obtained. Respective blot data were obtained from three independent experiments. (**B**) Immunofluorescent staining showed the subcellular distribution of p65 in nuclei and cytoplasm. Blue and green stains indicate nuclei and p65, respectively. Independent experiments were performed at least three times with similar results (magnification. 400×). (**C**) Intracellular ROS levels were determined by image analysis of DCFDA-loaded cells on a fluorescence microscope (magnification. 200X). (**D**) The fluorescence intensity (Ex: 485 nm per Em: 530 nm) of each cell lysate was then measured. Intracellular ROS levels are expressed as fold change in fluorescence intensity. Each value represents the means ± S.E. of four independent experiments. ** *p* < 0.01 vs. control and # *p* < 0.05 vs. AngII alone.

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
