# Peer review of "Sauchinone Protects Renal Mesangial Cell Dysfunction against Angiotensin II by Improving Renal Fibrosis and Inflammation"

_ijms, 2020, doi:10.3390/ijms21197003_

Round 1

Reviewer 1 Report

Yoon et al. examined the effect of Sauchinone on the Ang II-induced cell proliferation, ECMs and inflammatory components. The experiments were well conducted. However, there are some concern to reach the conclusion.

Major

  1. The authors mentioned that the results of this study are relevant to the pathogenesis of DN in the part of Introduction and Discussion. However, the overactivation of renin-angiotensin system mimicking by Ang II incubation is not a specific mechanism for DN. Thus, the reviewer thinks that the authors should more broadly discuss the pathogenesis of the mesangial proliferative disease, including IgA nephropathy or RAS activated kidney diseases.
  2. There is no information regarding the provider and dose of Ang II.

Minor

  1. In Figure 2B and 2C, are there any quantitative data? Also, “Sauchinone(uM)” may be missing in Figure 2B.
  2. In Figure 4B and 5B, the author should provide the sample size examined.

Reviewer 2 Report

This manuscript demonstrates that when added to in vitro cultures of human mesangial cells stimulated with ANGII, Sauchinone decreases the expression of fibrotic and inflammatory proteins.

Some minor comments:

-How was the dose of ANGII chosen?

-How were the doses of Sauchinone chosen?

-In methods:

details of primary antibodies (including catolog number and dilutions) would be helpful.

more information about cell culture would be helpful (starting from page 9 line 293), including the culture plates used and the number of cells seeded.

-On page 8 line 219, mesangial cells are not the main cause of interstitial fibrosis, but definitely contribute to glomerular sclerosis.

-These data are a great starting point to investigate the effect of Sauchinone on mesagnial cells, but the work lacks some in vivo confirmation. Is this treatment effective after the onset of disease (in all cases cells were pre-treated with Sauchinone)? And in an in vivo model of diabetes, is this treatment useful? Without this, the conclusion is a little overstated.

Round 2

Reviewer 1 Report

The revision was well done.